# Challenges and Opportunities in COPD Management in Latin America: A Review of Inhalation Therapies and Advanced Drug Delivery Systems

**DOI:** 10.3390/pharmaceutics16101318

**Published:** 2024-10-11

**Authors:** Juan S. Izquierdo-Condoy, Camila Salazar-Santoliva, Daniel Salazar-Duque, Yorlenis-Del-Carmen Palacio-Dávila, Juan M. Hernández-Londoño, Rafael Orozco-Gonzalez, María-Silvana Rodríguez-Sánchez, Valentina Marín-Bedoya, Valentina Loaiza-Guevara

**Affiliations:** 1One Health Research Group, Universidad de las Americas, Quito 170137, Ecuador; 2Facultad de Medicina, Fundación Universitaria Autónoma de Las Américas, Pereira 660001, Colombia; 3Facultad de Medicina, Universidad de Cartagena, Cartagena 130001, Colombia; 4Facultad de Medicina, Universidad Alexander von Humboldt, Armenia 630008, Colombia; 5Facultad de Medicina, Unidad Central del Valle del Cauca, Cali 763022, Colombia; 6Facultad de Medicina, Universidad de Manizales, Manizales 170008, Colombia

**Keywords:** chronic obstructive pulmonary disease, COPD management, inhalation therapies, drug delivery systems, nanoformulations, Latin America

## Abstract

Chronic obstructive pulmonary disease (COPD) is a leading cause of morbidity and mortality worldwide, particularly in low- and middle-income countries, where it poses a significant burden. In Latin America, the estimated prevalence of COPD is notably high, but the management and treatment of the disease have progressed slowly. This review examines the current status of inhalation therapy for COPD in Latin America, focusing on pharmacological therapies, inhalation devices, and the potential of advanced drug delivery systems. Pharmacological management predominantly relies on inhaled bronchodilators and corticosteroids, though access to these therapies varies considerably across the region. Inhalation devices, such as metered-dose inhalers (MDIs) and dry powder inhalers (DPIs), play a critical role in effective treatment delivery. However, their usage is often compromised by incorrect technique, low adherence, and limited availability, especially for DPIs. Emerging technologies, including nanoformulations, represent a promising frontier for the treatment of COPD by improving drug delivery and reducing side effects. However, significant barriers, such as high development costs and inadequate infrastructure, hinder their widespread adoption in the region. This review highlights the need for a multifaceted approach to enhance COPD management in Latin America, including optimizing access to existing inhalation therapies, strengthening healthcare infrastructure, improving provider training, and engaging patients in treatment decisions. Overcoming these challenges is crucial to improving COPD outcomes across the region.

## 1. Introduction

Chronic obstructive pulmonary disease (COPD) is a complex and heterogeneous respiratory condition characterized by persistent and progressive airflow obstruction. It affects millions of people worldwide and is one of the leading causes of global morbidity and mortality [1,2]. It is estimated that more than 380 million people suffer from COPD globally, representing approximately 5% to 10% of the adult population [3]. In 2019, COPD was the third leading cause of death worldwide, with 3.23 million deaths recorded [4]. This figure is expected to rise due to population aging and continued exposure to risk factors such as smoking, air pollution, and biomass fuel use, particularly in developing regions [4].

COPD is especially prevalent in low- and middle-income countries, where it imposes a significant burden on both individuals and healthcare systems [3,4]. In Latin America, the prevalence of COPD is around 8.9%, with higher rates observed among men, smokers, and ex-smokers. Prevalence in adults over 40 years of age ranges from 6% to 20% [5]. Prolonged exposure to irritants like cigarette smoke and air pollution is the primary risk factor. COPD not only diminishes patients’ quality of life but also imposes a substantial financial burden due to frequent exacerbations, many of which require hospitalization [6].

Countries such as Mexico, Brazil, and Colombia report particularly high prevalence rates, while in Ecuador, COPD ranks among the top ten causes of disability and is one of the top twenty causes of adult mortality [7]. The lack of access to early diagnosis and adequate treatment in many regions exacerbates the burden of the disease. The impact of COPD in Latin America is significant in terms of both public health and the economy [6,8]. The high mortality and disability associated with COPD have placed a significant strain on the region’s health systems [9]. Additionally, high rates of underdiagnosis and limited access to specialized treatments worsen the situation, reducing the quality of life for patients and increasing the costs associated with managing advanced complications [10].

The management of COPD includes the use of inhalation devices for drug delivery, which allows for the direct and effective administration of medications to the lungs. However, the effectiveness of inhalation therapies can be compromised by factors such as inhalation technique and accessibility to devices [10,11,12]. In this context, the introduction of new advanced drug delivery systems, including nanoformulations, presents an emerging opportunity to improve the treatment of various pathologies, including COPD [13,14]. However, while the outlook is promising, the actual implementation and potential action points for COPD treatment and management in regions such as Latin America remain unclear.

Therefore, this review aims to analyze the current state of treatment based on inhalation therapies and their innovative alternatives, such as nanoparticles, for COPD patients in Latin America.

## 2. Pharmacological Therapy for COPD

The pharmacological management of COPD primarily involves the use of bronchodilators, inhaled corticosteroids, and, in some cases, combination therapies [15]. These medications aim to alleviate symptoms, reduce the frequency and severity of exacerbations, and improve patients’ quality of life [16]. Below is a detailed description of the main classes of drugs used in COPD, along with their benefits and limitations.

### 2.1. Short-Acting Beta-2 Agonists (SABAs) and Long-Acting Agonists (LABAs)

Short-acting beta-2 agonists (SABAs), such as salbutamol and terbutaline, are first-line treatments in the management of COPD [17]. These medications provide rapid symptom relief by relaxing the smooth muscles of the airways, thereby facilitating breathing during acute episodes of dyspnea. The quick action of SABAs makes them ideal for immediate symptom relief, particularly in emergency situations or during exacerbations [17,18].

Conversely, long-acting beta-2 agonists (LABAs), such as formoterol and salmeterol, are used for long-term symptom control [19]. These drugs have a duration of action of approximately 12 h, allowing for sustained symptom management and a reduction in the frequency of exacerbations. However, LABAs do not affect the underlying progression of the disease and should be used with caution due to potential adverse effects, such as tremors, palpitations, and disturbances in heart rhythm. Despite their effectiveness in managing symptoms, the long-term use of LABAs requires careful monitoring to avoid significant side effects [19,20,21].

### 2.2. Long-Acting Muscarinic Antagonists (LAMAs)

Long-acting muscarinic antagonists (LAMAs), such as tiotropium and umeclidinium, are another essential class of drugs used in COPD management [22]. These medications work by blocking muscarinic receptors in the airways, leading to the relaxation of bronchial smooth muscle and a reduction in mucus production [21,23]. LAMAs effectively improve respiratory capacity and reduce the frequency of severe exacerbations, thereby contributing to a better quality of life for COPD patients [24].

A significant advantage of LAMAs is their favorable safety profile. The inhalational route minimizes systemic absorption, thereby reducing the risk of serious side effects. However, some patients may experience dry mouth or a bitter taste, which can affect adherence to treatment [25]. Additionally, LAMAs have been shown to enhance the effectiveness of pulmonary rehabilitation programs, making them a preferred choice for the long-term management of COPD [26,27].

### 2.3. Inhaled Corticosteroids (ICS)

Inhaled corticosteroids (ICS), such as fluticasone and budesonide, play a crucial role in managing COPD, particularly in patients with a history of frequent exacerbations [28]. These medications reduce airway inflammation, helping to prevent exacerbations and improve lung function when used in combination with LABAs.

The use of ICS is particularly indicated in COPD patients with a high blood eosinophil count or a history of concomitant asthma [2,29]. However, ICS are not without risks. Long-term use of these drugs may increase the risk of respiratory infections, including pneumonia. Therefore, the decision to use ICS should be carefully considered, weighing both the potential benefits and risks [30].

### 2.4. The Drawbacks of Drugs Used in COPD

The drugs used in COPD offer various benefits but also have limitations that should be considered in clinical management. SABAs and LABAs are effective for symptomatic relief and long-term control, but they do not alter disease progression and may have systemic side effects. LAMAs provide a favorable safety profile with a significant impact on reducing exacerbations, though they may cause minor adverse effects such as dry mouth. ICS are critical for preventing exacerbations in patients with specific inflammatory features, but prolonged use requires caution due to the risk of pneumonia [31,32,33]. The combination of a LABA with a LAMA compared to monotherapy reduces exacerbations and improves lung function (as measured by FEV1) in patients with moderate to severe COPD [34], although it does not influence the rate of lung deterioration or mortality. On the other hand, an increased risk of pneumonia has been associated with LABA/ICS dual therapy compared to LAMA therapy [35]. Findings on triple therapy indicate that the use of LABA + LAMA + ICS reduces exacerbations but increases the risk of pneumonia compared to dual LABA + LAMA therapy [36].

Furthermore, several factors may influence the effectiveness of inhaled medications in COPD patients. For instance, infections caused by bacteria, viruses, or both can exacerbate COPD symptoms, compromising the effectiveness of standard COPD treatment [37,38]. The choice of antibiotics can significantly impact clinical outcomes, especially in severe cases. COPD patients with an infectious phenotype may require a personalized therapeutic approach, including short- or long-term antibiotic use in addition to their usual COPD treatment [39].

Similarly, coexisting Cystic Fibrosis can worsen COPD symptoms even with treatment [40]. However, recent findings suggest that CFTR modulators, such as Ivacaftor (a CFTR enhancer), show promise in treating acquired CFTR dysfunction in COPD patients, particularly those with chronic bronchitis [41,42].

It is also important to note that external factors, such as the role of healthcare professionals, are crucial to ensuring the success of inhalation therapies. All healthcare providers should be proficient in teaching proper inhalation techniques [43]. Trainers must possess the necessary knowledge to effectively train patients in these skills, as evidence suggests that direct instruction from a pulmonologist significantly improves the development of proper inhalation technique [44,45]. This, in turn, enhances patient adherence to treatment, leading to improved clinical outcomes for COPD patients [43,46].

## 3. Inhalation Devices in COPD Treatment

The use of inhalation devices is essential in COPD management, allowing direct delivery of medications to the airways with minimal systemic side effects [47,48,49]. Various processes, starting with physiological mechanisms such as mucociliary clearance and phagocytosis by alveolar macrophages [50,51,52,53], combined with pharmacokinetic processes specific to the pulmonary environment, include drug dissolution in the lungs, drug deposition influenced by factors such as particle size, device type, and inhalation rate, as well as drug absorption and retention, which are especially dependent on the drug’s liposolubility [48,54,55]. Elimination characteristics, including ciliary movement, alveolar metabolic enzymes like superoxide dismutase, and clearance through systemic circulation [47,48,55], make inhaled drugs a promising alternative for COPD, though challenges must be addressed to ensure their effectiveness.

In this context, many of the processes that define the effect of inhaled treatments are related to the choice of the appropriate inhalation device, which in COPD patients depends on several factors, such as the type of medication, the patient’s ability to use the device correctly, the patient’s lung capacity, and the specific characteristics of the disease [56]. Currently, the main devices used in inhalation therapy for COPD patients include metered-dose inhalers (MDIs), dry powder inhalers (DPIs), nebulizers, and the new fine mist inhaler (SMI).

### 3.1. Metered-Dose Inhalers

MDIs are portable devices designed with a dosing valve that delivers a precise dose of medication encapsulated in aerosol form. They are widely used due to their convenience and portability. However, their effectiveness depends on precise coordination between device activation and inhalation, which can be challenging for some patients, especially those with physical or cognitive limitations [57]. Additionally, these devices have been shown to achieve deposition rates in lung tissue from 53% to 8% [58]. To improve the efficacy of MDIs and reduce the need for coordination between inhalation and activation, holding chambers are often used, which help optimize pulmonary drug deposition and reduce deposition in the oropharynx. Breath-activated metered-dose inhalers (baMDIs) have also been developed to eliminate coordination issues. These can be activated with low flow rates and provide better drug deposition; however, the range of drugs available for baMDIs remains very limited [12].

### 3.2. Dry Powder Inhalers

DPIs store medication without a propellant and are activated by the patient’s inspiration, eliminating the need for coordination between activation and inhalation. They are ideal for patients who can generate adequate inspiratory flow but may be less effective in those with compromised breathing capacity [59]. DPIs can be single-dose or multi-dose. However, the effectiveness of DPIs can be affected by factors such as ambient humidity, which can alter the stability of the powder. Although efforts have been made to improve the functionality of DPIs, their high cost presents a significant limitation, particularly in resource-limited settings. In terms of lung deposition, an important variation from 69% to only 7% was described from DPIs [58].

### 3.3. Nebulizers

Nebulizers convert a liquid solution into a fine aerosol inhaled through a mask or mouthpiece. They are especially useful for patients who have difficulty using MDIs or DPIs, such as children, the elderly, or patients with physical limitations [60]. While nebulizers are effective for administering medications in acute situations or severely compromised patients, they are less convenient for daily use due to their size, need for a power source, and long administration time [16]. Nebulizers also require regular maintenance to prevent contamination and ensure optimal operation. The main mechanisms of nebulizers include ultrasonic, vibratory mesh, and jet nebulizers, the latter being the most common in clinical settings [60,61]. However, despite their advantages, nebulizers achieve relatively low tissue deposition rates.

### 3.4. Fine Mist Inhalers

The fine mist inhaler (SMI) has been proposed as an innovative, next-generation alternative that does not use propellants and can deliver multiple doses of drugs. The characteristics of the inhalable aerosol, such as prolonged formation, make SMIs a superior alternative to other devices like MDIs [12]. However, the limited dosed volume (15 μL) presents a barrier to administering high-quality inhalable drugs [62]. Additionally, studies on the use of tiotropium in Respimat^®^ inhalers highlight concerns about the safety of the drug in patients with cardiovascular risks [12,63]. In addition, specifically, Respimat^®^ with different flow rates showed important lung deposition rates between 39.2% and 67% [58].

Table 1 provides a comparative overview of different inhalation devices used for treating COPD, detailing their features, advantages, and potential clinical.

## 4. Advanced Drug Delivery Systems

Recent research into advanced drug delivery systems has created new opportunities to improve COPD treatment. Nanoformulations, including nanoparticles, liposomes, and micelles, represent significant advances that offer the potential to transform disease management by providing more precise and effective delivery and reducing the range of adverse drug effects [14,64].

### 4.1. Nanoparticles and Liposomes

Nanoparticles and liposomes are drug delivery systems that encapsulate active ingredients, protecting them from premature degradation and facilitating controlled release at the target site. These systems allow for greater penetration into affected tissues, improving treatment efficacy and reducing the necessary dose, thereby minimizing side effects [13,65].

In COPD, nanoparticles are proposed to enhance the delivery of anti-inflammatory drugs and bronchodilators directly to the lungs, potentially resulting in more effective control of chronic inflammation and airflow obstruction. Liposomes have been shown to be effective in encapsulating drugs such as amikacin, used to treat chronic lung infections in COPD patients [66].

### 4.2. Micelles and Other Emerging Nanoformulations

Micelles and other emerging nanoformulations, such as nanoemulsions and nanocrystals, are designed to improve the solubility and bioavailability of drugs, particularly hydrophobic ones [67]. These technologies allow for sustained and targeted drug release, significantly improving treatment outcomes in COPD patients.

Micelles, for example, are supramolecular structures formed by the aggregation of surfactants that can encapsulate hydrophobic drugs in their core, protecting them from degradation and improving their transport to affected areas [68]. Nanoemulsions, on the other hand, are dispersions of two immiscible liquids stabilized by surfactants, which can improve the bioavailability and absorption of drugs administered via the pulmonary route [69].

### 4.3. Applications of Nanoformulations in the Treatment of COPD

Nanoformulations have the potential to revolutionize COPD treatment by overcoming many limitations associated with conventional therapies. By allowing more precise and efficient drug delivery, these technologies can reduce the necessary dose, improve treatment adherence, and decrease the frequency of exacerbations [13].

In vivo studies have shown that organic polymeric nanoparticles are more effective than metal nanoparticles in treating COPD. Al2O3 nanoparticles suppressed PTPN6 and STAT3 phosphorylation, protecting the lungs from inflammation. ZnO nanoparticles increased tumor suppressor protein expression in COPD patient samples. Nanoparticles targeting damaged elastin with controlled doxycycline release reduced MMP activity in rat lungs for up to 4 weeks [70,71,72]. Additionally, cationic polymeric nanoparticles effectively delivered miRNAs, while solid lipids reduced oxidative stress and inflammation in COPD mouse models. These studies highlight the potential of nanoparticles to provide sustained drug release, overcoming airway defenses and allowing therapeutic doses to be maintained in the lungs for prolonged periods with less systemic toxicity [14].

One of the most promising applications of nanoformulations is the delivery of inhaled corticosteroids via nanoparticles, which could significantly reduce the risk of side effects such as pneumonia [73,74]. Additionally, encapsulation of bronchodilators in nanoparticles could allow for sustained drug release, improving symptom control over a longer period and reducing the need for frequent dosing [74].

## 5. Inhalation Therapy in COPD Patients in Latin America

The management and treatment of COPD in Latin America has progressed slowly, with limited availability of specific evidence on inhalation therapy in this region. Most studies focus on local analyses, reflecting the diversity in treatment practices across different countries (Table 2).

### 5.1. Pharmacological Therapy

Regarding inhalation drug therapy for COPD, the available data show variations between countries. In Argentina and Peru, the combination of LABAs with ICS is among the most commonly reported strategies [75,88]. In contrast, in Colombia, large population studies have documented the use of long-acting β2-adrenergic agonists, long-acting muscarinic antagonists, and short-acting muscarinic antagonists, especially ipratropium bromide, as well as short-acting β2-agonists like salbutamol. Combinations of short-acting muscarinic antagonists with SABAs, and LABAs with a LAMA, are reported to a lesser extent [81,82,87].

In Brazil, LABAs, together with inhaled corticosteroids and short-acting β2-adrenergic agonists, are the most commonly used, according to studies conducted at the hospital in Bahia [83]. However, other studies have shown that only between 16.7% and 47% of COPD patients in countries such as Brazil, Argentina, Colombia, Venezuela, and Uruguay receive adequate treatment for COPD [84,86,89].

An additional study by Casas et al. found that combinations of inhaled corticosteroids, long-acting muscarinic antagonists, and LABAs, as well as the combination of ICS with LABAs, were the most commonly used treatments in Argentina, Chile, Guatemala, Mexico, and Uruguay [87].

### 5.2. Inhalation Devices

Regarding inhalation devices, pressurized metered-dose inhalers have been the most studied in Ecuador [76,77], although they are not necessarily the most commonly used. In Colombia, metered-dose inhalers with and without spacers are also the most used and studied, followed by single- and multi-dose dry powder inhalers and the fine mist inhaler [78,79,80,81].

Interestingly, a large portion of the literature has focused on exploring errors in the application of inhalation therapy and adherence to treatment in COPD patients in Latin America. However, most of these studies overlook several characteristics of inhalation therapy, such as the specific inhaled medications used. In Ecuador and Colombia, error rates in the use of metered-dose inhalers have been reported to range from 83.02% to 85% [76,77], while multi-dose dry powder inhalers showed an error rate of 16.2%. Fine mist inhalers and multi-dose dry powder inhalers of the turbuhaler type have presented very high percentages of correct inhalation technique (approximately 95%) [80]. Additionally, data from Brazil indicate adherence rates to inhalation therapy were 67.3% [85].

### 5.3. Implementation of Advanced Management Systems

With the evolution of pharmaceutical technology, advanced inhalation therapy delivery systems and nanoformulations have emerged as promising strategies for optimizing COPD management. However, integrating these technologies into clinical practice in Latin America presents various complexities that require in-depth analysis. In Latin America, the implementation of systems such as metered-dose inhalers with improved respirable particulate technology, dry powder inhalers with flow-optimized systems, and fine mist inhalers has been uneven [90,91]. Countries such as Colombia and Brazil have shown interest in adopting fine mist inhalers and dry powder inhalation technologies with improved lung deposition rates [80,92], but limited infrastructure and high costs have slowed their widespread adoption [90].

Available information indicates that while DPIs are accessible in some countries in Latin America, patients face significant challenges in obtaining them, primarily due to high costs [75]. The development and implementation of DPI technologies in the region are hindered by several barriers, including economic, regulatory, and technical obstacles [93]. The high costs of constructing advanced equipment, such as spray-drying and freeze-drying systems, impede the production of affordable DPI formulations suited to the region’s needs [94]. Moreover, limited investment in local pharmaceutical research and development (R&D) has resulted in a strong reliance on imports, further inflating the cost of introducing DPIs. Regulatory complexities and a lack of streamlined approval processes across the region contribute to delays in DPI adoption, limiting their widespread availability for conditions like COPD.

Addressing these challenges requires a coordinated approach involving collaboration between governments, academic institutions, and private industries to develop the necessary infrastructure for local DPI production. Public-private partnerships should be promoted to increase funding for affordable DPI development and support technology transfer, fostering innovation and cost-effective manufacturing [95]. Streamlining regulatory processes across Latin America and collaborating with international organizations could accelerate the approval of generic DPIs, reducing costs. Additionally, targeted government subsidies and pricing agreements with pharmaceutical companies could further enhance the affordability of DPIs, improving patient access to these essential treatments.

### 5.4. Integrating Nanoformulations into Clinical Practice

Nanoformulations have demonstrated substantial advantages, including controlled drug release, improved stability of the active ingredient, and reduced systemic side effects. However, in Latin America, the integration of nanoformulations into clinical practice is still in its early stages and faces significant barriers. Although the clinical use of nanoformulations has yet to result in drugs directly targeting COPD [96], their promising future remains distant for the Latin American region, which currently lacks adequate infrastructure for the production and evaluation of nanoformulations [97]. This deficiency limits their market availability and is compounded by significant knowledge gaps around access to inhalation therapies and COPD patient management, further complicating the clinical outlook for nanoformulations.

The implementation of advanced drug delivery systems in clinical practice still faces substantial challenges. The costs associated with the development and production of nanoformulations are often prohibitive, and the infrastructure required for their effective integration may be limited in certain regions [13,98]. In Latin America, despite the considerable therapeutic potential of nanoformulations for treating COPD, large-scale adoption necessitates significant investment in advanced equipment, such as spray-drying and freeze-drying systems, as well as highly skilled personnel [99].

A phased implementation strategy offers a practical approach to addressing these challenges. Initially, efforts should focus on reinforcing existing healthcare systems. Once these systems are sufficiently strengthened, nanoformulations can be introduced in more specialized medical facilities, particularly in urban areas where the necessary infrastructure and expertise are already established.

Collaboration between regional regulatory bodies and international organizations will be critical in streamlining the approval processes for nanotechnology-based medicines. Additionally, encouraging partnerships among governments, universities, and private industries across Latin America is key to securing funding and fostering innovation for creating nanoformulations tailored to the region’s unique health challenges.

By prioritizing innovation, affordability, and accessibility, these collaborations can ensure that nanomedicine reaches all socioeconomic groups, ultimately improving public health outcomes throughout the region.

## 6. Discussion

Given the increasing incidence of COPD patients, and despite the global trend over the last few decades toward deeper exploration of COPD and the use of inhalation therapies [100], this focus appears to be largely neglected in Latin American countries, as evidenced by this study. Colombia is likely the only country in the region where sufficient data are available to clarify the details of inhalation therapy in these patients [78,79,80,81,82].

Although the management and treatment of COPD have primarily focused on symptom control, it is well established that effective management reduces morbidity and mortality, leading to fewer exacerbations, hospitalizations, and overall complications. Evidence highlights the role of nitrative stress in the inflammatory process in COPD, with elevated alveolar nitric oxide concentration (CAlv) observed in patients. Studies suggest that CAlv, a marker for small airway dysfunction, is significantly correlated with lung diffusion capacity and airway obstruction in COPD [101]. This marker could serve as a useful tool in assessing the extent of inflammation and guiding personalized treatment strategies. Monitoring CAlv levels may help optimize inhalation therapies by identifying patients at higher risk of exacerbations or progressive lung function decline due to excessive nitric oxide production in peripheral airways.

Effective COPD management, especially in regions like Latin America, will require integrating such biomarkers with traditional pulmonary function assessments to provide a more comprehensive approach to patient care. This would enhance early detection of small airway dysfunction and potentially reduce long-term morbidity and mortality by adjusting treatments based on individual inflammatory profiles [102]. Despite the identification of studies in Latin America focused on evaluating inhalation techniques in countries like Ecuador and Colombia [76,77,80], or on treatment adherence in Brazil [85], the literature remains insufficient. Moreover, access to and use of inhaled therapies and devices are limited, with little to no data available from countries such as Ecuador, Venezuela, Bolivia, and Paraguay.

The choice of therapy for COPD patients depends on several factors, as each type of inhalation device has its advantages and disadvantages. The selection of a device should be based on an individualized assessment of the patient’s needs and abilities [103,104]. MDIs are compact and accessible but require a precise technique. DPIs are easy to use but may be ineffective for patients with low inspiratory flow. Nebulizers provide effective delivery for patients with severe breathing difficulties, but their size and maintenance requirements make them less practical for routine use [12,60,105]. Additionally, this choice must align with the availability of formulations and the patient’s ability to access these devices. In the region, this is compounded by deficiencies in access to treatments, such as long-acting muscarinic antagonists, as evidenced in Argentina [75].

The present and future management of COPD in Latin America will require joint efforts from key stakeholders, including healthcare providers, academics, patients, and government entities. For example, although errors in administration techniques have been identified [76,77,80], there are also shortcomings among clinicians in the region [75]. Therefore, increasing training for healthcare personnel and, subsequently, for patients is essential to improving control rates and successful COPD management in the region. Healthcare providers in the region are also encouraged to actively involve patients in decision-making, particularly in the selection of inhalation devices. Research shows that when patients participate in choosing their inhaler, they make fewer errors and exhibit better adherence to treatment. This collaborative approach not only improves their technique but also strengthens their commitment to long-term inhaler use, ultimately enhancing treatment outcomes [100,105,106]. An individualized evaluation approach, considering the patient’s inhalation capacity, cognitive function, availability of devices, and continuity in access to inhalation systems—factors that should be supported by the healthcare system or insurance providers—is crucial [107].

The integration of new technologies and pulmonary aerosol delivery represents a promising frontier in pharmaceuticals. Technologies such as lipid-based carriers and nanoparticles are being developed with the potential not only to treat respiratory diseases locally but also to deliver drugs systemically across the extensive lung surface, characterized by minimal barriers to the bloodstream [14]. Although lipid-based solid microparticles, such as liposomes, offer significant therapeutic advantages due to their low density and high porosity, nanostructured lipid carriers may have higher priority in the developing context of Latin America. These technologies offer formulation options that can be integrated into widely available inhalation devices in the region, such as nebulizers, MDIs, and DPIs, thereby improving the delivery of anti-inflammatories, bronchodilators, and antibiotics. This approach increases the bioavailability of insoluble hydrophobic drugs and allows for higher dosing.

However, the implementation of these advanced systems in Latin America faces several challenges, as has been evident with other innovative drugs [108]. These challenges include the high costs of development and production, the lack of adequate infrastructure to support widespread use, and the need for specialized training for healthcare professionals. The growing burden of COPD in Latin America may not necessarily require the adoption of advanced nanotechnology systems, at least in the short term. While nanoparticle-based drug delivery offers potential benefits, these technologies present significant challenges in terms of cost, infrastructure, and regulatory approval. Instead, more practical and cost-effective strategies could significantly improve outcomes for COPD patients in the region by optimizing existing treatments and strengthening healthcare systems.

A key area for improvement is enhancing healthcare infrastructure to ensure widespread access to existing inhalation therapies [109]. Traditional treatments, such as MDIs, DPIs, and nebulizers, have proven to be effective when properly implemented. However, many regions in Latin America, particularly rural and under-resourced areas, lack sufficient access to these basic medical devices. By improving the availability and distribution of inhaler systems, the region can achieve substantial health benefits without relying on complex nanotechnological solutions. Additionally, investments in local pharmaceutical manufacturing capabilities can further enhance access to these treatments, reducing dependence on costly imports [110].

This can be complemented by implementing healthcare strategies focused on educating healthcare personnel on the correct use of inhalers, which will, in turn, improve the education they provide to COPD patients. Furthermore, preventive healthcare efforts that address key risk factors, such as biomass exposure and cigarette smoking, along with improvements in public health initiatives, including vaccination programs and education on early detection of respiratory diseases, could have a significant impact. These strategies are both feasible and sustainable, providing immediate and long-term benefits to public health.

## 7. Conclusions

The management and treatment of COPD in Latin America face multiple challenges, characterized by diverse treatment practices and limited availability of specific evidence on inhalation therapies. There is considerable variability between countries in terms of the adoption and access to these therapies. For instance, countries like Colombia have shown progress in the use of various combinations of inhalers, while in others, the implementation of appropriate treatments is insufficient, leaving a high percentage of patients without the necessary care.

The effectiveness of inhalation devices in the region is compromised by high rates of errors in inhalation techniques and limited adherence to treatment, as reported in studies conducted in Ecuador, Colombia, and Brazil. This situation underscores the urgent need to improve education and training for both patients and healthcare professionals on the proper use of these devices.

Furthermore, implementing advanced drug delivery systems, including nanotechnology-based therapies, presents a promising opportunity to optimize COPD treatment. However, their integration into clinical practice in Latin America faces significant obstacles, including high development costs, inadequate infrastructure, and a lack of specialized training. These barriers limit access to innovative therapies, worsening the burden of COPD in the region.

To enhance COPD management in Latin America, a comprehensive approach is essential. This should involve optimizing access to current inhalation therapies, strengthening healthcare infrastructure, improving healthcare provider training, and actively engaging patients in treatment decisions to boost adherence. Additionally, practical strategies—such as expanding access to essential treatments, reinforcing public health initiatives, and investing in local pharmaceutical manufacturing—should take precedence over the immediate adoption of advanced nanotechnologies. By addressing these foundational challenges, substantial improvements in COPD outcomes can be realized.

## Figures and Tables

**Table 1 pharmaceutics-16-01318-t001:** Comparison of Inhalation Devices for COPD Management.

Device	Features	Advantages	Potential Clinical Applications
Metered-Dose Inhalers (MDIs)	- Delivers aerosolized medication through a pressurized canister. - Requires hand–lung coordination for proper activation and inhalation.	- Compact and portable. - Most treatment options available in this format. - Reproducible dose. - Suitable for emergency situations due to ease of use with low inspiratory flow requirements. - Less expensive than most other inhalers.	- Suitable for patients who can manage coordination. - Often prescribed for COPD patients requiring portable and immediate relief. - Commonly used for regular COPD management. - Effective in emergency situations requiring fast administration. - Can be used with spacers to improve coordination in patients with reduced hand–lung coordination.
Dry Powder Inhalers (DPIs)	- Propellant-free. - Activated by patient’s breath. - Available as single-dose, multi-dose blister, and reservoir-based systems.	- Overcomes coordination issues with MDIs. - Compact and portable. - Dose counters for multi-dose devices. - Ideal for patients with adequate inspiratory flow. - Short treatment time. - Available for most treatments.	- Ideal for patients with COPD who can generate adequate inspiratory flow. - Suitable for stable COPD management. - Less effective in patients with severe airflow limitation, especially during exacerbations. - Useful for patients with difficulty coordinating MDIs. - Limited in patients with low inspiratory flow.
Nebulizers	- Converts liquid medication into an aerosol for inhalation. - Requires a mask or mouthpiece. - Various types: air-jet, ultrasonic, vibrating-mesh.	- Delivers high doses of medication. - Ideal for patients with severe COPD or physical limitations. - Can be used without complex coordination or high inspiratory flow.	- Ideal for elderly or severely compromised COPD patients. - Commonly used in hospitals for acute exacerbations or for those unable to use other devices. - Commonly used in acute settings for severe COPD exacerbations. - Suitable for patients unable to use other devices due to physical limitations.
Soft Mist Inhalers (SMIs)	- Produces a slow-moving mist through a spring-loaded mechanism without the need for propellants. - Low inspiratory flow requirements.	- Reduces coordination issues compared to MDIs. - Portable multi-dose device. - High fine particle fraction leads to better lung deposition. - Low oropharyngeal deposition. - Suitable for use in children.	- Suitable for COPD patients requiring efficient drug delivery with minimal coordination. - Ideal for patients with cardiovascular risk concerns. - Effective in patients with coordination difficulties and reduced inspiratory flow. - Suitable for delivering bronchodilators and other COPD medications.

**Table 2 pharmaceutics-16-01318-t002:** Overview of Manuscripts Evaluating Inhalation Therapies in Latin America.

Author	Country	Study Design	Number of Participants	Main Outcomes	Inhaler Devices Assessed	Drugs Evaluated
Penizzotto M., 2024 [75]	Argentina	Cross-sectional	89	Limited access to recommended treatments; LABA/ICS most used due to lack of availability of LAMA and LABA/LAMA combinations	Not specified	LABA/ICS
Fernández M., 2019 [76]	Ecuador	Cross-sectional	64	Percentage of patients who do not perform a correct technique using pMDIs: 85% made errors using pMDIs; most common error: failure to exhale fully; correct technique reduced exacerbations (OR 0.56)	pMDIs	Not specified
Saraguro B., 2021 [77]	Ecuador	Cross-sectional	121	Determine the level and prevalence of adherence to inhaler use in patients with asthma and COPD. Poor adherence in asthmatics was 83.33%, and in COPD 13.33%.	pMDIs, DPIs	Not specified
Castrillón-Spitia J., 2023 [78]	Colombia	Cross-sectional	104	Describe the technique of using metered-dose inhalers and dry powder inhalers. The step most frequently skipped was connecting the inhaler to the inhalochamber (55.3%), while the least followed was waiting one minute between doses (9.6%).	pMDIs, DPIs	Not specified
Rojas M., 2022 [79]	Colombia	Cross-sectional	337	Validate the FSI-10 instrument and assess the degree of satisfaction with inhaler devices in COPD patients; 66% of patients who receive pharmacological treatment with pMDIs are somewhat satisfied, and only 28% are very satisfied with the device.	pMDIs	Not specified
Montes J., 2023 [80]	Colombia	Cross-sectional	80	Evaluate the technique of using inhaled drugs and describe errors in inhaler technique. Incorrect technique was found in 48.7%. The most frequent error was the failure to perform pre-inspiratory exhalation.	pMDIs, DPIs, SMIs	Not specified
Valladales-Restrepo L., 2023 [81]	Colombia	Cross-sectional	500	Establish the adherence of patients diagnosed with COPD; 56.6% had good adherence.	pMDIs, DPIs, SMIs	LABAs, LAMAs, SABAs, SAMAs, ICS
Machado-Duque M., 2023 [82]	Colombia	Cross-sectional	9476	Determine the trends of COPD medication. The most frequently used therapy was SAMAs (39.9%), followed by SABAs (31.6%). Combination of SABA + SAMA + ICS (8.7%), and 9.3% of patients received LAMA, 1.9% were on U-LABA monotherapy, and 14.2% were on a LABA + LAMA combination; 6.8% were on triple therapy (LABA + LAMA + ICS).	Not specified	LAMAs, SABAs, LABA/LAMA combinations
Ribeiro C., 2019 [83]	Brazil	Cross-sectional	383	Describe COPD pharmacological treatment patterns. LABAs (47.5%), ICS (44.9%), SABAs (33.7%), SAMAs (9.7%), LAMAs (9.7%), and methylxanthines (9.4%).	Not specified	LABAs, ICS, SABAs, SAMAs, LAMAs, Methylxanthines
Ávila G., 2023 [84]	Brazil	Cross-sectional	636	Estimate the prevalence of treatments used for the management of COPD; 49.4% of diagnosed patients reported using treatment, followed by oxygen therapy and physical therapy.	Not specified	Not specified
Alcantara de Moreira A., 2021 [85]	Brazil	Cohort	333	Evaluate the association between adherence to treatment and mortality among COPD patients; 67.3% were adherent to treatment; non-adherence was associated with increased mortality.	Not specified	Not specified
Nascimento O., 2007 [86]	Brazil	Cross-sectional	144	Evaluate the proportion of COPD patients who had never been diagnosed and determine if diagnosed COPD patients were receiving appropriate treatment; 87.5% had never been diagnosed; 82.3% of COPD patients were not receiving any pharmacological treatment.	Not specified	Not specified
Casas A., 2018 [87]	Argentina, Chile, Colombia, Costa Rica, Guatemala, Mexico, Uruguay	Cross-sectional	795	Evaluate the type of medication used and adherence. ICS/LAMA/LABA (32.9%), ICS/LABA (27.7%), LABA/LAMA (11.3%), SABA or SAMA (7.9%), LABA (6.4%), LAMA (5.8%), and ICS (4.3%). The use of long-acting bronchodilators showed the highest adherence (>50%).	pMDIs, DPIs, SMIs	ICS, LABA, LAMA, SABA, SAMA, LABA/LAMA combinations
Guerreros A., 2018 [88]	Peru	Cross-sectional	196	Describe the clinical characteristics of patients with COPD. The most frequently used therapy was LABA/ICS (31.1%).	Not specified	LABA, ICS
Jardim J.R., 2017 [89]	Argentina, Colombia, Venezuela, Uruguay	Cross-sectional	309	Assess respiratory medications used, factors predicting treatment, and patterns of corticosteroid use; 79.4% used respiratory medications: bronchodilators (64.7%), corticosteroids (37.6%), bronchodilators + corticosteroids (25.6%); 79% of patients with a prior diagnosis were using corticosteroids.	Not specified	ICS

## Data Availability

Not applicable.

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
