# Peer review of "Challenges and Opportunities in COPD Management in Latin America: A Review of Inhalation Therapies and Advanced Drug Delivery Systems"

_pharmaceutics, 2024, doi:10.3390/pharmaceutics16101318_

Round 1

Reviewer 1 Report

Comments and Suggestions for Authors

1.A table that highlights the features, advantages, and potential clinical applications of devices would be helpful.

2.Improving activity levels is a significant challenge in COPD treatment. Therefore, inhalation therapy strategies that induce behavioral changes are crucial. Please include considerations on inhalation therapy from the perspective of prognosis.

3. Adherence is a major issue in COPD inhalation therapy. Therefore, it is important to consider both inhalation therapy and personalized medicine.

4. The importance of controlling peripheral airway inflammation in COPD has been highlighted, as noted in the reference(DOI; 10.1088/1752-7155/7/4/046002). Please include considerations on peripheral airway inflammation and inhalation therapy strategies.

Author Response

Dear reviewers, we greatly appreciate your work in reviewing our manuscript. In this regard, we have worked hard to respond to each of your comments in search of the best version of our manuscript.

Below you will find a point-by-point response to each of your requests, and the revised version of our manuscript with the corrections in a different font color for easy identification.

Reviewer 1

Comments and Suggestions for Authors

1.A table that highlights the features, advantages, and potential clinical applications of devices would be helpful.

Thank you very much for your suggestion. We have added a table that summarizes the features and main applications of inhalation devices.

2.Improving activity levels is a significant challenge in COPD treatment. Therefore, inhalation therapy strategies that induce behavioral changes are crucial. Please include considerations on inhalation therapy from the perspective of prognosis.

Thank you very much for your valuable comment. We have corrected and improved the text within the manuscript around inahalatory therapy considerations for COPD patients.

3.Adherence is a major issue in COPD inhalation therapy. Therefore, it is important to consider both inhalation therapy and personalized medicine.

Thank you very much for your valuable comment. We have increased the wording on characteristics and factors related to adherence to treatment and its importance in the management of COPD patients.

4. The importance of controlling peripheral airway inflammation in COPD has been highlighted, as noted in the reference(DOI; 10.1088/1752-7155/7/4/046002). Please include considerations on peripheral airway inflammation and inhalation therapy strategies.

Thank you for your valuable suggestion. We have corrected and included further discussion on peripheral airway inflammation considerations for the selection of inhaled therapy alternatives.

Reviewer 2 Report

Comments and Suggestions for Authors

The manuscript demonstrated the Challenges and Opportunities in COPD Management in Latin America. The title is very interesting; however, the content is not focused.

How would the authors adopt the emerging nanotechnological systems in Latin America, where the accessibility of currently available devices is limited for the management COPD.  The nanotechnological systems are still under investigation even in the developed countries.

A lot of DPI devices with a wide range of resistance are available on market. Appropriate selection of DPIs for the patients may improve the ongoing difficulties.  The appropriate training for the health professionals, improvement of patient adherence in using the devices might help. 

The abstract needs to be rewritten focusing the ongoing challenges and their possible solutions.   

Section 2.4: Could be replaced with "the drawbacks of drug used in COPD" as there are no doubts on their quality/potency. Appropriate training of healthcare professionals, patients adherence is required to ensure the best outcomes. Lung infections associated with COPD and CF should be  included here as it creates a huge problem in the treatment processes. 

Section 3: The first para needs to be rewritten to address this section. all types of devices ie. MDIs, DPIs and nebulizers may be presented as a) MDIs, b) DPIs and so on; and then discuss the merits and demerits of the devices with a view to select the appropriate device for a patients depending on the respiratory conditions. 

line 140-143; not clear

line 144-146 is not linked with the previous line

Section 4: Advanced drug delivery technologies demonstrated in this article are not new and most nanobased formulations are under development. If the Latin Americans are unable to afford the currently available devices (MDI, DPIs or nebulizers); how are these advanced technologies be adopted there? 

Section 5.3: needs more explanation focusing on the barriers on the development and implementation of DPI technologies in Latin America and how can the authors proposal would help solve the problems. DPIs are available everywhere in the world. Are they expensive in Latin America?

5.4: this section needs more insights on how the authors would think to implement in Latin America as the  affordability of the currently available products is under difficulty?

Discussion is very short. It should be demonstrated the possible ways of improving the ongoing difficulties in Latin America rather than adopting the nanotechnological systems. 

Conclusion: should be rewritten based on the purpose of this review article. 

Comments on the Quality of English Language

as above

Author Response

Point by point letter

Dear reviewers, we greatly appreciate your work in reviewing our manuscript. In this regard, we have worked hard to respond to each of your comments in search of the best version of our manuscript.

Below you will find a point-by-point response to each of your requests, and the revised version of our manuscript with the corrections in a different font color for easy identification.

Reviewer 2

Comments and Suggestions for Authors

The manuscript demonstrated the Challenges and Opportunities in COPD Management in Latin America. The title is very interesting; however, the content is not focused.

How would the authors adopt the emerging nanotechnological systems in Latin America, where the accessibility of currently available devices is limited for the management COPD.  The nanotechnological systems are still under investigation even in the developed countries.

-A lot of DPI devices with a wide range of resistance are available on market. Appropriate selection of DPIs for the patients may improve the ongoing difficulties.  The appropriate training for the health professionals, improvement of patient adherence in using the devices might help. 

Thank you for your valuable comment. We have increased the information on the importance of adherence to treatment in the context of inhalation therapy and have included a table summarizing the characteristics and applicability of the different types of inhalers

-The abstract needs to be rewritten focusing the ongoing challenges and their possible solutions.   

Thank you for your valuable comment. The manuscript has been revised and corrected based on all your suggestions and comments. So the abstract has been corrected along with the manuscript.

-Section 2.4: Could be replaced with "the drawbacks of drug used in COPD" as there are no doubts on their quality/potency. Appropriate training of healthcare professionals, patients adherence is required to ensure the best outcomes. Lung infections associated with COPD and CF should be  included here as it creates a huge problem in the treatment processes. 

Thank you for your valuable comment. We have corrected the subtitle of section 2.4, in addition, we have increased and improved the wording around factors that may jeopardize the effectiveness of inhalation therapies in COPD patients.

-Section 3: The first para needs to be rewritten to address this section. all types of devices ie. MDIs, DPIs and nebulizers may be presented as a) MDIs, b) DPIs and so on; and then discuss the merits and demerits of the devices with a view to select the appropriate device for a patients depending on the respiratory conditions. 

Thank you for your suggestion. We have rewritten and corrected the first paragraph of section 3, furthermore we have structured the information of section 3 in sub sections, finally we have added a table (Table 1) that synthesizes the information about the different inhalation devices available for COPD patients in the Latin American region.

-line 140-143; not clear

Thank you for your comment. The information on lines 140 - 143 has been corrected to clarify your understanding.

-line 144-146 is not linked with the previous line

Thank you for your comment. The information on lines 144 - 146 has been corrected to clarify your understanding.

-Section 4: Advanced drug delivery technologies demonstrated in this article are not new and most nanobased formulations are under development. If the Latin Americans are unable to afford the currently available devices (MDI, DPIs or nebulizers); how are these advanced technologies be adopted there? 

Thank you for your comment. The authors agree with the reviewer's point of view, the adoption of advanced technologies such as nanoparticles in the region is presented on a limited and unclear landscape, in that context we have increased the information in the manuscript around how the region could overcome the barriers for the implementation of nanoparticle-based treatments in COPD patients.

-Section 5.3: needs more explanation focusing on the barriers on the development and implementation of DPI technologies in Latin America and how can the authors proposal would help solve the problems. DPIs are available everywhere in the world. Are they expensive in Latin America?

Thank you for your valuable comment. Unfortunately, official information about access and availability of DPIs in Latin America is limited, however, unofficial information demonstrates wide access barriers to these devices in COPD patients in several Latin American countries, in this sense a proposal has been provided to overcome these barriers considering the importance and benefits associated with the management with DPIs.

-5.4: this section needs more insights on how the authors would think to implement in Latin America as the affordability of the currently available products is under difficulty?

Thank you for your comment. We understand the difficulties that are included in the region regarding the integration of emerging therapies such as nanoformulations, in that context, we have included proposals that seek to resolve these difficulties in the region within the manuscript.

 -Discussion is very short. It should be demonstrated the possible ways of improving the ongoing difficulties in Latin America rather than adopting the nanotechnological systems. 

Thank you for your comment. The discussion of the manuscript has been expanded on several important points in relation to the objectives of the review.

-Conclusion: should be rewritten based on the purpose of this review article. 

Thank you for your suggestion. The conclusions of the manuscript have been corrected according to the corrections and purposes of the manuscript.

Round 2

Reviewer 1 Report

Comments and Suggestions for Authors

The revised manuscript is well written.

Reviewer 2 Report

Comments and Suggestions for Authors

The authors addressed all concerns raised with my satisfaction.